# The Molecular Genetics of Gordon Syndrome

**DOI:** 10.3390/genes10120986

**Published:** 2019-11-29

**Authors:** Holly Mabillard, John A. Sayer

**Affiliations:** 1Renal Services, The Newcastle Hospitals NHS Foundation Trust, Newcastle upon Tyne NE7 7DN, UK; hollymabillard@gmail.com; 2Translational and Clinical Research Institute, Faculty of Medical Sciences, Institute of Genetic Medicine, Newcastle University, International Centre for Life, Central Parkway, Newcastle upon Tyne NE1 3BZ, UK; 3NIHR Newcastle Biomedical Research Centre, Newcastle University, Newcastle upon Tyne NE4 5PL, UK

**Keywords:** Gordon syndrome, tubulopathy, genetics, *WNK1*, *WNK4*, *KLHL3*, *CUL3*

## Abstract

Gordon syndrome is a rare inherited monogenic form of hypertension, which is associated with hyperkalaemia and metabolic acidosis. Since the recognition of this predominantly autosomal dominant condition in the 1960s, the study of families with Gordon syndrome has revealed four genes *WNK1*, *WNK4*, *KLHL3*, and *CUL3* to be implicated in its pathogenesis after a phenotype–genotype correlation was realised. The encoded proteins Kelch-like 3 and Cullin 3 interact to form a ring-like complex to ubiquitinate WNK-kinase 4, which, in normal circumstances, interacts with the sodium chloride co-symporter (NCC), the epithelial sodium channel (ENaC), and the renal outer medullary potassium channel (ROMK) in an inhibitory manner to maintain normokalaemia and normotension. WNK-kinase 1 has an inhibitory action on WNK-kinase 4. Mutations in *WNK1*, *WNK4*, *KLHL3*, and *CUL3* all result in the accumulation of WNK-kinase 4 and subsequent hypertension, hyperkalaemia, and metabolic acidosis. This review explains the clinical aspects, disease mechanisms, and molecular genetics of Gordon syndrome.

## 1. Introduction

Gordon syndrome, otherwise known as Pseudohypoaldosteronism type 2 (https://omim.org/phenotypicSeries/PS145260) or familial hyperkalaemia and hypertension syndrome, is a rare inherited form of low-renin hypertension associated with hyperkalaemia and metabolic acidosis (specifically a type IV renal tubular acidosis) [1,2,3,4]. Whilst the glomerular filtration rate (GFR) is usually preserved, it can manifest as a particularly severe phenotype with hyperkalaemia (serum potassium reaching as high as 8 to 9 mmol/L), resulting in periodic paralysis. Gordon syndrome is an interesting outlier amongst other forms of familial hypertension because of its hypoaldosterone hyperkalaemia phenotype in contrast with other forms of monogenic hypertension, which have normo- or hypokalaemia, thus making serum potassium a useful discriminator of the condition [5]. The condition was first described in Australia in the 1960s and was named after Richard Gordon after he realised the inherited phenotype by tracking several Australian families with the condition [6]. Pedigrees of this syndrome have since been discovered all over the world, but the discovery of a distinct phenotype–genotype correlation was made when French pedigrees were noted to have much milder manifestations of the condition [7,8]. To date, age of diagnosis has varied from (in one researched cohort) 7 months to 39 years and no relationship has been found between the severity of biochemistry and age or increase in blood pressure. The phenotypic variability includes the degree of sensitivity to a thiazide diuretic, which is the precise treatment for this condition and, along with observed pattern of inheritance and linkage studies, indicates heterogeneity [9]. No genetic linkage has been found with *SLC12A3*, which encodes the thiazide sensitive NaCl co-symporter (NCC), in which ‘activating’ mutations would be predicted to cause the disease phenotype, given the sensitivity to thiazide diuretics. Most families follow an autosomal dominant pattern of inheritance, but recent research has shown some phenotypes of Gordon syndrome to be inherited in an autosomal recessive manner [10]. Initially, through linkage studies, two loci were identified, the first on chromosome 1 (1q31-q42) (PHA2A), which is still unsolved, and chromosome 17 (17p11-q21) (PHA2B). Following this, an intronic deletion was detected on chromosome 12 (12p13), which occurred within *WNK1* (PHA2C), which, along with *WNK2/3/4*, encodes a family of WNK-kinases [11,12]. *WNK4* mutations (PHA2B) were also subsequently identified in Gordon syndrome patients. These exonic missense mutations occurred within a unique amino acid sequence, ‘the acidic motif’ of *WNK4*, and resulted in charge-changing substitutions but did not affect the kinase activity of *WNK4*. This ‘acidic motif’ has only made sense since the discovery that the WNKs bind to Kelch-like 3 through the acidic motif [12,13]. Lifton’s group, using whole exome sequencing, identified two further genes implicated in Gordon syndrome, *KLHL3* (5q31), which encodes Kelch-like 3 (PHA2D), which was also identified by a French group [14], and *CUL3* (2q36), which encodes Cullin 3 (PHA2E) [15]. Both of these proteins have been found to be involved in the endosomal degradation of *WNK4* [16]. Gordon syndrome is therefore a group of related disorders, with at least four known monogenic aetiologies. This review will describe the clinical presentations and disease mechanisms of known genes implicated in this condition, *WNK1*, *WNK4*, *KLHL3*, and *CUL3*.

## 2. Clinical Presentations

Over 25% of the adult population in the developed world have hypertension [17], which is an independent risk factor for myocardial infarction, cerebrovascular disease, and renal failure, and a leading cause of morbidity and mortality globally. The kidneys are a major contributing factor to the development of hypertension in most cases [18,19]. In total, 5% to 10% of patients with hypertension have a secondary cause [20]. Secondary causes should be suspected in patients who have early onset hypertension (<30 years old) in the absence of common risk factors, such as obesity. Factors, such as resistant hypertension (BP >140/90 despite three or more anti-hypertensive drugs), severe hypertension (BP >180/110 or hypertensive emergency), sudden BP increase in a previously stable patient, presence of target organ damage, or non-dipping or reverse dipping during 24 h ambulatory BP monitoring [21], should also be considered. Essential hypertension is a polygenic disease with over 280 genetic variants found to be associated with risk of high blood pressure [22,23,24]. Monogenic forms of secondary hypertension, however, should be suspected in a family history of the disease coupled with either low or high serum potassium levels in the presence of supressed renin secretion and metabolic alkalosis or acidosis [25]. However, mild hypertension and the absence of electrolyte abnormalities do not exclude hereditary conditions and therefore hormonal studies should be coupled with genetic testing if a secondary or monogenic cause is suspected.

Of the monogenic forms of secondary hypertension, most result from increased sodium transport along the aldosterone-sensitive distal nephron [26]. Most monogenic forms manifest with hypokalaemia, often because the epithelial sodium channel (ENaC) is involved; these mechanisms are displayed in Figure 1. Gordon syndrome is also known as Pseudohypoaldosteronism type II as it is the only form of monogenic hypertension that manifests as hypoaldosteronism because of increased volume expansion, through increased sodium chloride reabsorption in the distal convoluted tubule (DCT), which decreases luminal sodium flow in the more distal nephron and makes ENaC less functional. Gordon syndrome may manifest with hyperkalaemia partly due to a similar way to the downregulation of ENaC by amiloride. Aside from hyperkalaemia and metabolic acidosis, a mild hyperchloraemia and calcium leak can be seen (especially with *WNK4* mutations), which can present as hypercalciuria, hypocalcaemia, low bone mineral density, and renal stones, and these findings can precede hypertension [27]. This is evidence that *WNK4* positively interacts with TRPV5, a calcium resorptive channel [28]. Table 1 highlights the genotype–phenotype correlations in Gordon syndrome. The final indicator of a possible diagnosis of Gordon syndrome is the increased sensitivity to a thiazide diuretic. This can even cause hypotension, but at an appropriate dose will correct hypertension and all electrolyte abnormalities [8] and therefore once treatment is established, risk of cardiovascular disease, cerebrovascular disease, and renal failure is unlikely. Although hypertension does increase risk of cerebrovascular accidents, in contrast to glucocorticoid remedial hyperaldosteronism, Gordon syndrome does not have a strong association with haemorrhagic stroke. Table 2 summarises the differential diagnoses of Gordon syndrome.

Gordon syndrome is a form of monogenic hypertension with a hypoaldosterone phenotype. Other forms of monogenic hypertension, including Liddle syndrome, congenital adrenal hyperplasia, glucocorticoid remedial hyperaldosteronism, and syndrome of apparent mineralocorticoid excess, manifest with hypokalaemia due to hyperstimulation of ENaC in the principal cells of the collecting duct in the nephron. Aldosterone interacts with the mineralocorticoid receptor and translocates to the nucleus where transcription of ENaC subunits occur, which then populate the apical membrane of principal cells. ENaC removes cationic sodium from the lumen, which makes it electronegative. This promotes passive potassium secretion from the tubular cell into the lumen via apical ROMK channels. Liddle syndrome is caused by missense or frameshift mutations that disrupt the organization of the ENaC subunits. This disrupts ENaC from binding to Nedd4-2, a ubiquitin-protein ligase, which degrades ENaC, resulting in its increased apical membrane expression and subsequent hypokalaemia. Congenital adrenal hyperplasia secondary to mutations in *CYP21A2* (which encodes 21-hydroxylase) results in defective conversion of 17-hydroxyprogesterone to 11-deoxycortisol and so cortisol synthesis is reduced. Low cortisol increases ACTH secretion, which increases steroid intermediate synthesis, such as deoxycorticosterone (DOC). DOC has potent mineralocorticoid action, which stimulates the expression of ENaC, resulting in hypokalaemia. Glucocorticoid remedial hyperaldosteronism occurs due chimeric gene changes where the 5-prime regulatory sequences of *CYP11B1* are fused to the coding region of *CYP11B2*, which results in ectopic expression of aldosterone synthase in the zona fasciculata, resulting in constant stimulation of the mineralocorticoid receptor and subsequent increased ENaC expression and hypokalaemia. The syndrome of apparent mineralocorticoid excess is caused by a deficiency in 11-β-hydroxysteroid dehydrogenase, which results in a defect of the peripheral metabolism of cortisol to cortisone. Cortisone excessively cross-reacts at the mineralocorticoid receptor, thereby stimulating ENaC and hypokalaemia.

## 3. The Regulation of NCC

Gordon syndrome presents as a biochemical and phenotypic ‘mirror image’ of Gitelman syndrome, a salt wasting disease caused by inactivating mutations of *SLC12A3* encoding NCC. Most of the biochemical defects of Gordon syndrome are corrected by treatment with a thiazide diuretic, suggesting that *WNK* kinases regulate NCC, and this has been confirmed by in vitro studies [29,30]. *WNK1*, *WNK3*, and *WNK4* are all expressed in the kidney. WNK signalling in the kidney controls blood pressure and electrolyte homeostasis by managing two opposing aldosterone-controlled processes: NaCl reabsorption and potassium excretion in the distal nephron [31]. WNK signalling regulates the phosphorylation of and activities of cation-chloride co-transporters (CCCs), which include NCC (found in the DCT), KCC4 and NKCC2 (found in the TAL), ROMK, and ENaC (found in the DCT and CD). WNKs phosphorylate NCC, NKCC1, and NKCC2 via the phosphorylation and activation of SPAK and OSR1 [32,33,34,35,36]. Chloride is regulated via NCC influx and KCC efflux and this maintains transepithelial solute and water transport and volume regulation in addition to neuronal stimulation. Chloride depletion and cell shrinkage results in WNK phosphorylation of NCC, NKCC1, NKCC2, and KCC and the opposite occurs in chloride repletion. It can be concluded that intracellular chloride influences whether WNKs have an inhibitory or activatory effect on NCC [37,38,39]. As *WNK4* is the most sensitive to the chloride of the WNKs, this explains the early postulations in the literature that *WNK4* was initially inhibitory of NCC as chloride environments were an unrecognised confounding factor [40].

*WNK1* produces two isoforms: L-WNK1 (a longer form) and KS-WNK1 (a shorter kidney specific WNK1). KS-WNK1 lacks kinase activity and is only expressed in the distal nephron. Early in vitro cell line and *Xenopus* oocyte studies did not demonstrate a direct effect of *WNK1* on NCC, but *WNK1* was shown to abolish WNK4′s inhibitory effect on NCC and phosphorylate SPAK, suggesting an important role in NCC regulation [41,42,43,44]. More recently, a mouse model of a human *WNK1* mutation (large deletion of the first intron of *WNK1*) demonstrated a L-WNK1/SPAK pathway for NCC activation. These mice demonstrated a full Gordon syndrome phenotype with increased NCC and SPAK expression and phosphorylation in the DCT in addition to an increase in L-WNK1 in the DCT and CD with no change to KS-WNK1 expression [33,45]. Recent studies have shown that WNK4 becomes active in the presence of KS-WNK1 [46]. *WNK3* has a more minor role in the distal nephron and activated NCC in a kinase- and SPAK-dependent manner [47,48]. *WNK3* knock-out mice have a milder phenotype and show a slight blood pressure reduction with salt depletion. In the absence of *WNK3*, NCC phosphorylation is probably maintained by compensatory increased expression of L-WNK1 [49]. Interestingly, a kinase dead *WNK3* model is a strong NCC inhibitor, demonstrating a dominant negative mechanism where the absence of WNK3 has an opposing effect [47,50]. *WNK4* is necessary for phosphorylation and activation of NCC in in vivo mice studies via SPAK. When *WNK4* was inactivated, NCC expression and activity significantly reduced and a hypokalaemic metabolic alkalosis developed and angiotensin II no longer phosphorylated SPAK and NCC. The opposite occurred when *WNK4* was overexpressed in murine models, which resulted in reduced blood pressure, hypokalaemia, and hypocalciuria, essentially a Gitelman phenotype [51]. In contrast, other transgenic murine studies where *WNK4* was significantly overexpressed (two-fold higher levels of WNK4 kidney protein) produced increased phosphorylation of OSR1, SPAK, and NCC in the kidney, leading to a Gordon syndrome phenotype [52]. Clearly there is a balance of *WNK4* protein levels required for physiological NCC regulation (see Figure 2).

More recently, a deeper understanding of the WNK pathway’s involvement in ion transport has uncovered an additional mechanism for the generation of hyperkalaemia and metabolic acidosis in Gordon syndrome. *WNK4* has been shown to significantly inhibit Maxi-K channel (BK) activity and expression in a kinase-dependent manner. This is done via lysosomal degradation rather than clathrin-dependent endocytosis as WNKs do with other membrane proteins. This demonstrates a broader role of *WNK4* in potassium homeostasis [53]. Mice transgenic for a *WNK4* missense mutation (TgWnk4^PHA2^) seen in Gordon syndrome show activation of Pendrin activity (an anion exchange protein encoded by *SLC26A4*), suggesting *WNK4* is involved in Pendrin activation [53]. This theory has also been suggested in other studies [54,55]. Pendrin has been shown to work in concert with NCC to favour electroneutral NaCl reabsorption in distal nephron at the expense of ENaC suggesting that disordered regulation of Pendrin is also involved in the generation of hyperkalaemia in Gordon syndrome [56,57,58,59,60,61]. This is also validated in that decreased urinary potassium secretion in TgWnk4^PHA2^ mice is reversed by Pendrin deletion. When Pendrin is genetically ablated in TgWnk4^PHA2^ mice, hyperkalaemia resolves and this could be due to enhanced BK activity in intercalated cells as the BK-β-4 subunit is present in intercalated cells with the BK-α subunit in addition to L-WNK1, which activates BK [62]. Pendrin also has a role in the generation of metabolic acidosis in Gordon syndrome. Pendrin-mediated cellular Cl^−^/HCO3^−^ exchange and number of Pendrin-expressing cells significantly increase and genetic ablation of Pendrin completely corrects metabolic acidosis in TgWnk4^PHA2^ mice. This suggests that metabolic acidosis is, in fact, likely to be caused by distal nephron bicarbonate wasting in Gordon syndrome [63].

*WNK4* has been shown to be regulated by various other molecules that clearly influence WNK signalling. Osmotic stress and hypotonic low-chloride stimulation increase *WNK4* Ser575 phosphorylation via the p38MAPK-MK pathway, suggesting that this pathway might regulate WNK4 in an osmotic stress-dependent manner [64]. Other molecules that have been shown to stimulate elements of the WNK-SPAK pathway include aldosterone, angiotensin II, vasopressin, norepinephrine, insulin, and parathyroid hormone. Glucocorticoids, estrogen, progesterone, and prolactin certainly have a stimulatory effect on NCC; however, their exact mechanisms are currently unknown [65]. Interestingly, the extensively used organ transplant immunosuppressant, Ciclosporin, alters the WNK signalling pathway, and can cause hypertension, hyperkalaemia, and metabolic acidosis [66]. Another calcineurin inhibitor, Tacrolimus, also impairs NCC dephosphorylation, stimulating NCC [67]. The chemotherapeutic agent, Cisplatin, has an inhibitory effect on NCC by reducing NCC mRNA expression [68].

Wild-type *WNK4* phosphorylates and activates NCC via SPAK activation and mutant *WNK4* further phosphorylates and activates *WNK4* in a SPAK-dependent manner [51]. Wild-type *WNK3* has a more modest effect on NCC. WNK3 phosphorylates and activates NCC via SPAK and this is only done in a kinase-dependent manner. Kinase-dead *WNK3* acts in the opposite way and inhibits NCC [47,48,49,50]. Wild-type L-WNK1 phosphorylates NCC via SPAK and this pathway is exaggerated in mutant *WNK1* models even in the absence of *WNK4* [33,45].

## 4. Discovery of SPAK and OSR1

Further research into the WNK signal transduction pathway revealed two new molecules important in the pathway involving WNK isoforms and their role in blood pressure and electrolyte homeostasis: SPS1-related proline/alanine-rich kinase (SPAK) and oxidative stress-responsive kinase (OSR1). SPAK was initially identified as a novel kinase in a pancreatic β-cell PCR screen [69] and OSR1 was discovered in sequencing a large DNA region of human chromosome 3 during a search for tumour suppressor genes [70]. Phylogenetic tree analysis of all human kinase domains placed the WNK family closest to the STE20 protein kinase family, which includes SPAK and OSR, hinting that these kinases interact in some way [71]. Further work demonstrated that SPAK and OSR also interact with the aforementioned CCC membrane-associated proteins. These included KCC (K-Cl co-transporter), NKCC1 (Na-K-2Cl cotransporter), NKCC2, WNK4, and WNK1 with respect to SPAK and CLH3 (CIC-type chloride channel); and *WNK1*, *WNK2*, and *WNK4* for OSR1 [41,72,73,74,75]. WNKs phosphorylate and activate SPAK and OSR1, which subsequently phosphorylate the CCCs [31]. SPAK has three isoforms and FL-SPAK (full length) and KS-SPAK (kidney specific) have been seen to co-localize with NCC at the DCT and NKCC2 in the TAL, respectively, in immunofluorescence studies [73,76]. Shorter SPAK isoforms might have an inhibitory effect on NKCC2 as SPAK knock-out mice had an increase in NKCC2 activity and phosphorylation whilst this was reduced in SPAK knock-in mutants. Clearly, OSR1 does not compensate enough for the absence of SPAK in these models [77]. When OSR1 is inactivated, a phenotype similar to Bartter syndrome develops due to reduced phosphorylation and activation of NKCC2. Interestingly, when a double SPAK-OSR1 knock-out mouse was analysed, these was still significant phosphorylation of NKCC2, suggesting a third kinase is involved in this cascade [77]. Various mouse models have been created to assess the role of SPAK. In every model with defective or absent SPAK, NCC phosphorylation is substantially reduced and a Gitelman syndrome phenotype is observed, which demonstrates the importance of SPAK and OSR1 in the WNK cascade of CCC, especially NCC and NKCC2, activation [76,77,78].

More recently, several studies have demonstrated that the WNK-SPAK/OSR1 pathway components concentrate into large WNK bodies at discrete foci in the DCT in response to potassium balance changes. For example, during hypokalaemia, the diffuse distribution of WNK, SPAK, and OSR1 transition to form a punctate presence in the DCT, which have been seen in genetically modified mice [79]. When total body potassium balance is altered via dietary or genetic manipulation to the WNK-SPAK/OSR1 pathway, these puncta are exclusively present in this environment and almost entirely limited to the DCT and have been shown to require KS-WNK1 for their expression [76,80,81,82,83]. WNK1 forms DCT-specific particles during potassium loading or restriction. This ‘renal potassium switch’ has become fundamental in our understanding of the role of potassium intake in the development of hypertension via the WNK-SPAK-NCC pathway. For example, low potassium intake induces hyperpolarization of the early DCT cell membrane, which favours Cl^-^ efflux across the basolateral membrane and decreases intracellular Cl^-^ concentration and activates WNK-SPAK signalling. WNK4 forms heterodimers with ks-WNK1 and assembles with ribosomal protein 22 sub nuclear puncta and causes SPAK-induced NCC phosphorylation [64]. This WNK-WNK reaction requires a cysteine-rich hydrophobic motif within a unique N-terminal exon of KS-WNK4 and these WNK-WNK assemblies require aldosterone in high potassium environments [66,67].

Kidney-specific SPAK is highly abundant along the thick ascending loop of Henle (TAL) whereas full-length SPAK is more abundant along the DCT. SPAK knock-out animals show increased phosphorylation of NKCC2 in the TAL and decreased phosphorylation in the DCT. In mice, extracellular fluid depletion influences SPAK abundance to favour NaCl retention along both segments, which indicates that a SPAK isoform switch modulates sodium control across the distal nephron [78]. Mouse models confirm that OSR1 mainly activates NKCC2-mediated sodium transport along the TAL while SPAK mainly activates NCC along the DCT, but the kinases compensate for each other [84].

## 5. Mechanisms of Hyperkalaemia in Gordon Syndrome

Understanding the ion transport protein pattern is important to be able to understand how the WNK kinases, Kelch-like 3 and Cullin 3, regulate electrolyte homeostasis. Three views on the mechanism of hyperkalaemia in Gordon syndrome currently exist: Direct reduction on K^+^ secretion (theory one), a secondary deficiency in K^+^ secretion due to lack of chloride (Cl^−^) reaching ENaC (theory two), and lack of Na^+^ reaching ENaC for adequate exchange with K^+^ (theory three). Potassium is secreted mostly via ROMK channels so it would be expected that these channels would be depleted in Gordon syndrome. Chloride reabsorption via the paracellular pathways (including claudin channels) is increased, resulting in less negativity at the ENaC and subsequent depression of the ROMK transporter. Therefore, the chloride shunt would be increased in Gordon syndrome. This chloride shunt theory was that pre-treatment with mineralocorticoid, intravenous Na_2_SO_4_, and NaHCO_3_ could cause increased potassium excretion in Gordon syndrome but that sodium chloride was less able to do so. A chloride shunt proximal to the ENaC would result in less chloride delivery and subsequently less ENaC negativity, which would result in decreased potassium secretion [4]. Other studies have reinforced this theory, but mathematical modelling of the DCT does not demonstrate the need of this theory to explain Gordon syndrome. Finally, increased NCC activity results in a reduction in Na^+^ reaching ENaC for reabsorption, thereby depressing its action and resulting in hyperkalaemia. Figure 3 summarises the mechanisms of potassium homeostasis involving WNK-kinases and the impact of WNK-kinases on each of these protein transporters will now be discussed.

The impact of wild-type *WNK4* and wild-type *WNK1* on five tubular channels are demonstrated. Red arrows represent inhibitory effects and green arrows represent stimulatory effects. WNK4 causes increased paracellular chloride channel (claudin channel) permeability, which is exacerbated by mutant *WNK4* [34]. *WNK4* also inhibits the epithelial sodium channel (ENaC) by stimulating glucocorticoid-induced kinase 1 and the opposite is true for mutant WNK4 [31,85]. *WNK4* reduces plasma membrane renal outer medullary potassium channel (ROMK) and maxi-K channel (BK) abundance and these inhibitory effects are markedly increased by mutant WNK4 [34,53]. *WNK1* stimulates ENaC via phosphatidylinositol 3-kinase and it stimulates BK expression. As mutant WNK1 increases *WNK1* expression, ENaC and BK stimulation are exaggerated in Gordon syndrome [33,86,87,88]. *WNK1* inhibits ROMK activity and therefore its inhibition is also exaggerated by mutant WNK1 [89]. KS-WNK1 stimulates ENaC and inhibits WNK1′s stimulation of the same channel and its inhibition of ROMK [88,90,91]. Wild-type WNK4 regulates Pendrin, but mutant WNK4 stimulates Pendrin activity and surface expression [56,57,58,59,60,61].

### 5.1. Lack of Na^+^ Reaching ENaC for Adequate Exchange with K^+^

*WNK1* has been found to increase the activity of ENaC by activating phosphatidylinositol 3-kinase, which stimulates glucocorticoid-induced kinase 1 (a well-known ENaC-regulatory factor). Interestingly, this effect of *WNK1* on ENaC depends on an intact *WNK1* kinase domain but also requires an intact amino acid terminal domain (N-terminal of the kinase domain), but KS-WNK1 (kidney specific) also stimulates ENaC without both of these domains [33,86,87,88]. This implies a different ENaC-stimulating mechanism between *WNK1* and KS-WNK1. WNK4 does the opposite to ENaC activity; *WNK4* stimulates glucocorticoid-induced kinase 1, which inhibits ENaC. When expressed in *Xenopus* oocytes and in mice, ENaC activity is not inhibited by the Gordon syndrome mutant *WNK4* and subsequent increased ENaC activity is seen in the kidney and colon in these mutant models [31,85]. Now that ENaC involvement has been established, three observations justify why more than just ENaC is involved in the pathogenesis of Gordon syndrome. Firstly, thiazide diuretics correct both hyperkalaemia and hypertension in patients with Gordon syndrome, including in patients who have *WNK4* mutations and animal models with mutant Wnk4. Thiazide diuretics affect NCC and do not directly affect ENaC but do increase distal Na^+^ delivery where ENaC expression is abundant. Second, the increased ENaC activity in WNK4 D561A knock-in animals can be reversed with thiazide diuretics, which suggests that these effects on ENaC are not directly from WNK kinases; they must be secondary physiological effects. Finally, ENaC stimulation should result in hypokalaemia, not the hyperkalaemia we see in Gordon syndrome.

### 5.2. Direct Reduction on K^+^ Secretion

*WNK4* reduces plasma membrane ROMK abundance in the distal nephron and inhibits, through a kinase-independent mechanism, ROMK activity in vitro [85,90]. Its effects on ROMK are dependent on dynamin and involves clathrin-medicated endocytosis, which is not the case when compared with the *WNK4* effect on NCC [92]. *WNK1* inhibits ROMK as well but whether this depends on intact kinase activity is in conflict and this is done by dynamin-dependent endocytosis [89]. KS-WNK1 does not have a direct effect on ROMK, but it does block the effects of *WNK1* on this channel in addition to inhibiting NCC [90,91]. It has been shown that mutant *WNK4* additionally inhibits ROMK in comparison to wild-type *WNK4,* so this effect is more powerful, resulting in a reduced plasma membrane ROMK abundance [86]. Again, as for ENaC, now that we have established ROMK involvement in Gordon syndrome, it is clear that ROMK cannot be the only channel affected in Gordon syndrome due to the following three reasons. Firstly, treatment with a thiazide diuretic would not be expected to correct an ROMK defect [93,94], yet it corrects hyperkalaemia in Gordon syndrome. However, you would expect urinary sodium and water excretion to increase with thiazide diuretic use due to distal flow stimulating potassium secretion but serum potassium is lowered, despite urinary Na and water remaining unchanged, from baseline in Gordon syndrome subjects, suggesting further distal alternative channel involvement. Secondly, when non-chloride Na^+^ salts are infused in Gordon syndrome subjects, normal amounts of potassium are excreted, demonstrating intact potassium secretory apparatus and that, in Gordon syndrome, impaired potassium secretion is a result of the reduced (aldosterone sensitive) distal nephron voltage [4,95]. Thirdly, mutant *WNK4* transgenic animals do not appear to have a difference in ROMK abundance [93,94].

### 5.3. A Secondary Deficiency in K^+^ Secretion Due to Lack of Chloride Reaching ENaC

A distal nephron chloride shunt is also proposed as a mechanism of hyperkalaemia in Gordon syndrome. It has been suggested that an increased permeability of paracellular chloride channels in the distal nephron results in increased Cl^-^ reabsorption and a subsequently less negative gradient at ENaC, so it is depressed. ENaC depression results in decreased Na^+^ reabsorption and reduced potassium secretion, causing hypertension and hyperkalaemia [4]. Evidence that *WNK4* is involved in this process is supported in various ways. *WNK4* has been observed to colocalize with tight junctions where the paracellular chloride channels are [96]. Furthermore, when overexpressed in culture cells, *WNK4* has been shown to affect paracellular chloride permeability [34]. Also, mutant *WNK4* increased chloride permeability when compared to Na^+^ permeability and this required catalytic activity of *WNK4* (*WNK4* phosphorylated claudins 1–4 and kinase-inactive *WNK4* did not affect chloride channel permeability) [34]. However, as with the other channels, the ability of a thiazide diuretic alone to correct the hyperkalaemia is not consistent with dominant chloride channel involvement in Gordon syndrome as this directly interacts with NCC and has never been seen to alter the paracellular chloride channels. A further argument against chloride channel involvement is that WNK4 D561A knock-in mice show no difference in the measurement of paracellular permeability when compared with wild-type collecting ducts [94].

## 6. The Role of Kelch-like 3 and Cullin 3

More recently, mutations in two further genes have been identified, *KLHL3* and *CUL3*, which account for 80% of families with Gordon syndrome [15,97]. *KLHL3* mutations can be dominant or recessive whilst *CUL3* mutations are usually dominant or de novo [15]. Kelch-like 3 (encoded from *KLHL3*) contains three domains: A BACK domain, an N-terminal BTB domain, and a six-bladed β-propeller structure formed from Kelch-like repeats [98,99]. These Kelch propeller domains bind substrate proteins so that they can be ubiquitinated via their interaction with the BTB domain of Cullin 3 (encoded from *CUL3*). This interaction is termed the CUL3-KLHL3 E3 ligase complex (a ring-like complex) [99,100,101]. Ubiquitination via this complex not only targets proteins for degradation but also modulates protein activity, localisation, and interaction [102,103]. Kelch-like 3 is predominantly expressed in the apical DCT and collecting duct whilst Cullin 3 is expressed across all segments of the nephron, especially in the proximal tubule, when stained in mice kidneys [97]. This initially demonstrated the role of this complex in distal nephron electrolyte homeostasis.

Immunoprecipitation of Kelch-like 3 demonstrated its strong association with WNK isoforms in addition to Cullin 3. In total, 13 out of 15 dominant *KLHL3* mutations inhibited the binding of Kelch-like 3 to *WNK1* or Cullin 3 and when studies were performed in vitro, the recombinant wild-type CUL3-KLHL3 E3 ligase complex (but not the disease causing complex) ubiquitinated *WNK1* [102]. Furthermore, siRNA-mediated *CUL3* knockdown increased *WNK1* levels and its kinase activity in HeLa cells [103]. The *KLHL3* interaction site in *WNK1* is on a non-catalytic region, which is interestingly the equivalent site on WNK4, which encompasses mutated residues in Gordon syndrome patients. *WNK4* mutations (E562K and Q565E) and an equivalent *WNK1* fragment mutation (479–667), a deletion, prevents their ability to interact with Kelch-like 3 [103]. Knock-in mice carrying *KLHL3* disease causing mutations exhibited salt-sensitive hypertension, hyperkalaemia, and metabolic acidosis and revealed that both *WNK1* and *WNK4* were increased within the kidney due to impaired KLHL3-Cullin 3-mediated ubiquitination [104] Homozygous knockout *KLHL3* mice models also produced a Gordon syndrome phenotype with increased *WNK1* and *WNK4* levels in the kidney [105]. All of this provides strong evidence that the CUL3-KLHL3 E3 ligase complex regulates blood pressure via its ability to ubiquitinate WNK isoforms and that Gordon syndrome results in mutations that disrupt the formation of this complex’s ability to prevent WNK degradation and therefore stimulate renal salt retention by increased activation of NCC and NKCC2 by WNK isoforms in excess. Figure 4 summarises the pathogenic effects of *WNK4*, *KLHL3*, and *CUL3* mutations.

When Gordon syndrome-causing *KLHL3* mutations are examined, it is clear that they affect the binding of Kelch-like 3 with the acidic motif on WNK4-kinase in different ways. This is either by altering the electrostatic potential of the acidic domain-binding site or by disrupting the Kelch-like 3-acidic motif hydrogen bonds. Mutations buried inside the Kelch-like 3 domain either have little or no impact on the interaction, suggesting that some buried mutations probably disrupt the Kelch-like 3-acidic motif interaction by a different mechanism [106]. A Gordon syndrome mouse model has also been created with a *Klhl3* mutation in the BTB domain, which should aid a better understanding of the Kelch-like 3/Cullin 3 interaction and contribute to possible future novel antihypertensive drug targets [107].

Kelch-like 3 is highly expressed along the DCT and substantially expressed along the TAL and collecting duct. However, Cullin 3 is uniformly expressed along all nephron segments except for the glomerulus, with the highest levels in the proximal convoluted tubule [108,109]. The DCT is highly sensitive to Gordon syndrome mutations in *CUL3* because of the extremely high Kelch-like 3 expression there. The proportional reduction in Kelch-like 3 levels have significant consequences on ion transport in the DCT in comparison to other segments. *CUL3* knock-out models show significant defects along all nephron segments and also resulted in reductions on other tubular proteins, such as AQP2 and NKCC2, which override increased NCC expression, so a phenotype of salt wasting and polyuria develops rather than Gordon syndrome [109]. Interestingly, KS-CUL3 mice develop interstitial inflammation and fibrosis, in addition to a similar fibrotic picture when *CUL3* is deleted from the liver. Also, humans can develop papillary renal cell carcinoma with loss-of-function *CUL3* mutations. This shows that loss of *CUL3* activity is detrimental to cells and that KS-CUL3 mice could be a good candidate for experimental studies to slow CKD progression with *CUL3* activating molecules [109].

The phenotypic severity of Gordon syndrome varies according to different causative mutations (*CUL3* > recessive *KLHL3* > dominant *KLHL3* > *WNK4* > *WNK1*) [97]. Gordon syndrome subjects with *CUL3* mutations have the most severe phenotype and present at a younger age and had more severe hyperkalaemia, hypertension, and metabolic acidosis in addition to growth impairment. As many of these subjects had de novo *CUL3* mutations, impairment of reproductive fitness is likely [97]. Recently, the severe phenotype of *CUL3* mutations has been explained by the mutation’s effect on vascular function in addition to renal. Activation of RhoA, a key regulator of vascular tone, was shown in mutant *Cul3* mouse models in addition to acute blood pressure sensitivity to calcium channel blockers (Amlodipine) and altered aorta reactivity to phenylephrine and acetylcholine [110].

In wild-type conditions (left panel), *WNK4* can stimulate NCC trafficking to the plasma membrane of the DCT and its activation is achieved by phosphorylation of NCC by SPAK/OSR1, stimulating NaCl transport. *WNK1* and *WNK4* can be ubiquitinated by Kelch-like 3 and Cullin 3. *WNK4* is inhibited by *WNK1*. *WNK1* also acts via SPAK and increases the phosphorylation of NCC. In Gordon syndrome (right panel), the pink arrows represent an increased abundance of mutant *WNK1* and wild-type *WNK1* and *WNK4* and the green arrows (marked +) indicate increased expression of NCC on the apical membrane of the distal convoluted tubule (DCT), NKCC1 on the basolateral membrane of the collecting duct (CD), and NKCC2 on the apical membrane of the thick ascending loop of Henle (TAL). The red crosses signify the inability of *WNK4* to bind with the Cullin 3-RING ubiquitin ligase complex. Mutant *WNK4* overstimulates plasma membrane proteins NCC, NKCC1, and NKCC2 because mutant WNK4 is unable to bind and be ubiquitinated by Kelch-like 3 and Cullin 3; mutations in Kelch-like-3 or Cullin 3 lead to *WNK1* and *WNK4* accumulation because of a failure of the Cullin 3-RING ubiquitin ligase complex to form and ubiquitinate WNK [9,103]. Wild-type and mutant *WNK4* phosphorylates SPAK and OSR1 to activate NCC, NKCC1, and NKCC2. Mutations in *WNK1* are intronic deletions and do not affect the protein structure but lead to changes in the expression of a *WNK1* isoform, which can phosphorylate and stimulate NCC.

## 7. Conclusions

By studying monogenic forms of hypertension, such as Gordon syndrome, new molecular pathways that regulate blood pressure and electrolyte homeostasis have been discovered. Not only will this prove helpful for the development of new drug targets for hypertension, one of the leading causes of cardiovascular disease, but it may help us to identify and categorise new subgroups of patients with hypertension based on genotype–phenotype correlations. In the future, more bespoke anti-hypertensive treatment will hopefully become a reality because of a deeper understanding of the physiology of blood pressure regulation in the kidney.

## Figures and Tables

**Figure 1 genes-10-00986-f001:**
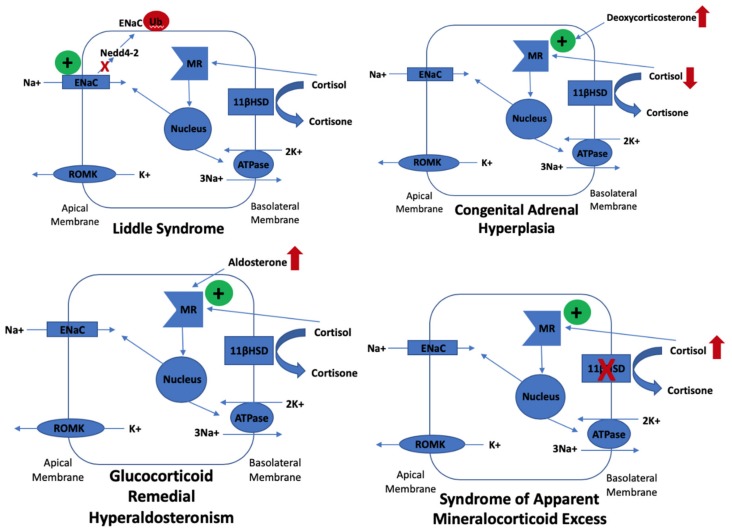
Molecular mechanisms underlying Gordon syndrome.

**Figure 2 genes-10-00986-f002:**
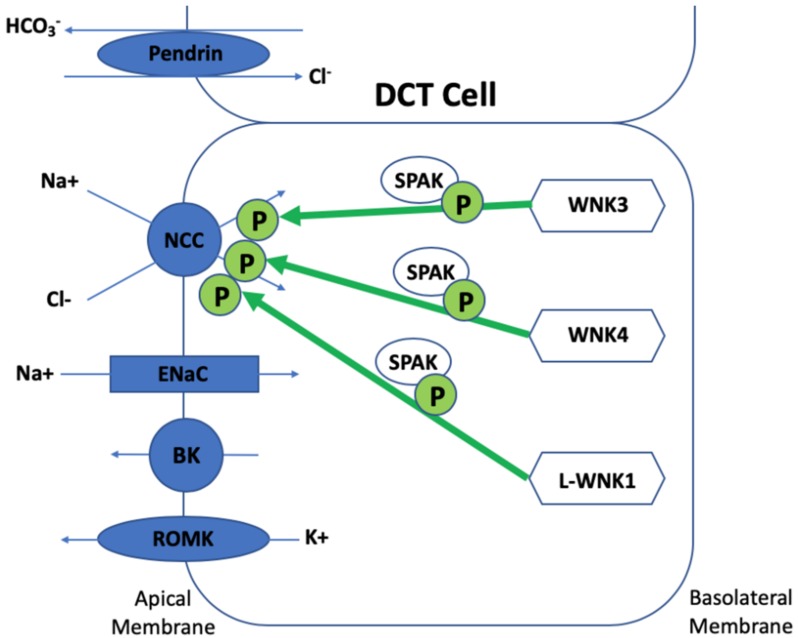
Regulation of NCC by WNKs.

**Figure 3 genes-10-00986-f003:**
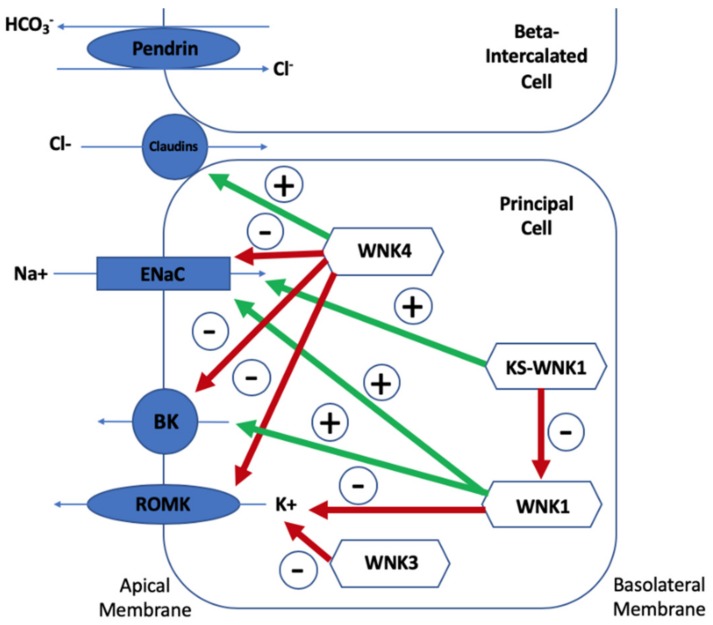
Mechanisms of potassium homeostasis in the β-intercalated and principal cells of the collecting duct.

**Figure 4 genes-10-00986-f004:**
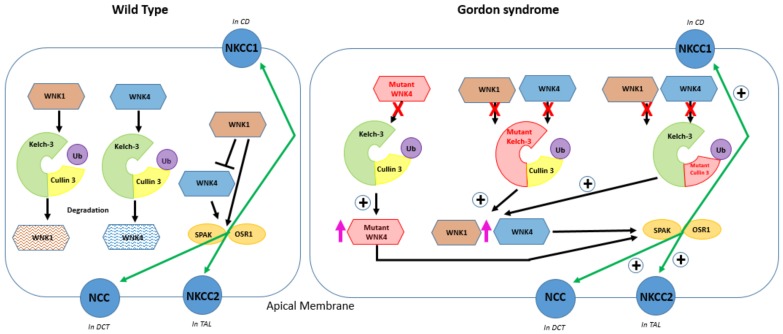
Summary of pathophysiology of Gordon syndrome in a tubular cell representative of different parts of the nephron.

**Table 1 genes-10-00986-t001:** Phenotype–genotype correlations in Gordon syndrome.

	*WNK1*	*WNK4*	*KLHL3*	*CUL3*
Hypertension	Least severe phenotype and metabolic disorder often precedes hypertension	Metabolic disorder often precedes hypertension	Recessive mutations are more severe and diagnosed at an earlier age than dominant mutations	Most severe phenotype. Presents at youngest age (>90% had hypertension <age 18.
Hyperkalaemia	Least severe	Yes	Dominant mutations had significantly higher serum K^+^ than recessive mutations	Most severe Presents at youngest age
Metabolic Acidosis	Least severe	Yes	Yes	Most severe
Other features		Hypercalciuria Hypocalcaemia Decreased bone mineral density Renal calcium stones		Fertility likely affected in de novo mutations. Growth impairment most likely

**Table 2 genes-10-00986-t002:** Differential diagnosis of Gordon syndrome.

Diagnosis	Genes/Loci	Gene Product	Inheritance	Reason for Hypertension	Other Features
Gordon Syndrome	1q31-q42 (Unknown gene) 17p11-q21 (*WNK4*) 12p13 (*WNK1*) 5q31 (*KLHL3*) 2q36 (*CUL3*)	Mutant WNK-kinase 1, WNK-kinase 4, Kelch-like 3 or Cullin 3	Dominant (can be recessive or *de novo*)	Excessive sodium reabsorption via NCC	Hyperkalaemia Hyperchloraemia Metabolic acidosis Very thiazide diuretic sensitive **Occasionally** Hypocalciuria Stones (calcium) Low BMD
Liddle Syndrome	16p12.1 (*SCNN1B* and *SCNN1G*) 12p13.31 (*SCNN1A*)	Faulty ENaC (β subunit SCNN1B, γ subunit SCNN1G or α subunit SCNN1A)	Dominant	Excessive sodium reabsorption by ENaC	Hyperkalaemia Metabolic acidosis **Occasionally** Hyperkalaemia Early renal failure Early onset stroke
Congenital Adrenal Hyperplasia	1p12 (*HSD3B2*) 6p21 (*CYP21A2*)	3-β-hydroxysteroid dehydrogenase 2 deficiency 21-hydroxylase deficiency	Recessive	Excessive ACTH to try to maintain cortisol which causes excess mineralocorticoid-like hormones which cross react with mineralocorticoid receptors	Hyperkalaemia Metabolic acidosis Hypocortisolism/high ACTH Ambiguous external genitalia Early development of secondary sex characteristics Short stature
Glucocorticoid Remedial Hyperaldosteronism	8q24 (*CYP11B1* & *CYP11B2*)	18-hydroxylase, 11-β hydroxylase hybrid gene causing aldosterone to be under ACTH control	Dominant	Inappropriately high aldosterone levels	Hyperkalaemia Metabolic acidosis **Occasionally** Hyperkalaemia Increased haemorrhagic stroke risk
Syndrome of Apparent Mineralocorticoid Excess	16q22 (*HSD11B2*)	11-β hydroxysteroid dehydrogenase (type 2) deficiency	Recessive	Defect in ability to metabolise cortisol to cortisone resulting in cortisol cross reactivity with the mineralocorticoid receptor	Hyperkalaemia Metabolic acidosis Hypernatraemia

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
