# Peer review of "The Molecular Genetics of Gordon Syndrome"

_genes, 2019, doi:10.3390/genes10120986_

Round 1
Reviewer 1 Report
The authors provide a well-written and comprehensive review on a rare form of hypertension, the Gordon syndrome.
I have a few amendments, that should be addressed:
1. In Table 2, the differential diagnosis of Gordon syndrome is given. Here, the data for Liddle syndrome (LS) are incomplete. LS cannot only be caused by mutations in the genes encoding the beta (SCNN1B) and gamma (SCNN1G) subunit, which are located on chromosome 16p12.2. Recent findings showed that also mutations in the gene encoding the alpha subunit (SCNN1A) are causally related to LS (Salih et al., 2017). As SCNN1A locates to chromosome 12p13.31, these informations should be included in the table.
2. I am missing the data from several novel manuscripts and the findings presented here should be included in the review, e.g.
López-Cayuqueo et al. A mouse model of pseudohypoaldosteronism type II reveals a novel mechanism of renal tubular acidosis. Kidney Int. 2018;94:514-523.
Lin et al. Generation and analysis of a mouse model of pseudohypoaldosteronism type II caused by KLHL3 mutation in BTB domain. FASEB J. 2019;33:1051-1061.
Wang et al. Unveiling the Distinct Mechanisms by which Disease-Causing Mutations in the Kelch Domain of KLHL3 Disrupt the Interaction with the Acidic Motif of WNK4 through Molecular Dynamics Simulation. Biochemistry. 2019;58:2105-2115.
Chen et al. WNK4 kinase is a physiological intracellular chloride sensor. Proc Natl Acad Sci U S A. 2019;116:4502-4507.
and several more from the last two years
Minor comments
1. Page 2, line 81: should it read… In contrast to this….?
2. Page 5, line 157: The authors mention the chloride shunt theory by Shambelan and coworkers but do not give the correct reference in line 161. It should be no. 4 instead of 54. Hence, the references have to be renumbered from no. 54 in the reference list and in the text.
3. References are given in inconsistent style and not always in complete (e.g. no.38) form.
Author Response
We wish to thank our peer reviewers for the extremely helpful comments on our paper ‘The molecular genetics of Gordon syndrome’.
Our changes are highlighted in yellow to demonstrate clearly what has been updated.
Below is an account of our response to reviewers comments.
In table 2, we have added the recommended genetic information on Liddle Syndrome including the gene mutations which encode the various ENaC subunits. We have incorporated more recent literature into our review including those suggested by our reviewers including research from the last 2-3 years. Some examples of updated references are reference numbers 41, 42, 48, 55, 64, 65, 66, 67, 68, 82, 83, 84, 86, 108 and 112 Page 2, line 81 (now line 89) ‘in contrast to this’ has been removed as we agree that this statement was not used in the correct context Page 5, line 57 (still line 157) the reference has been corrected so that this is no longer duplicated and the order of references have been adjusted In section 3 ‘The regulation of NCC’, we have re-written this section to reflect more recent literature and understanding of the positive regulation of NCC by WNK4 and the role of SPAK/OSR1 in this. Additionally, we have more clearly described the role of WNK3 and figure 2 and 3 have been updated accordingly. In section 4 ‘Discovery of SPAK and OSR1’, we have re-written this section to incorporate more recent literature and have more clearly outlined the function of SPAK and OSR1 and its role in NCC activation. In figures 1 (now 2) and 3 (now 4), we have removed the incorrect specification that the DCT cell is ‘DCT2’ in nature and clearly specified the different locations of the membrane transporters in figure 3. In section 1 (line 32-35) and 2 (line 84-89), we have put more emphasis on the hypoaldosterone phenotype of Gordon syndrome and that this differs from other monogenic forms of hypertension. In addition to this, we have added a figure (figure 1) to illustrate why ENaC causes potassium loss in other monogenic forms of hypertension as recommended. In table 2, we have changed ‘patients with Gordon syndrome manifest hypocalciuria’ to ‘hypercalciuria’ In section 1 (line 60-61) we have acknowledged and referenced (reference 14) that the French group also reported that a KLHL3 mutation causes GS. In line 223 (now 358), ENaC depression results in increased Na+ reabsorption has been changed to ‘decreased’ Na+ reabsorption. Information about distribution of WNKs, SPAK and OSR1 in the nephron has been added to section 4 (lines 244-270). Distribution of Kelch-3 and Cullin3 along the nephron has been added to section 6 (lines 455-467). Figures 1-3 (now 1-4) all now have specification of the apical and basolateral cell sides. In Figure 3 (now figure 4), the membrane transporter proteins have been moved to the appropriate membrane sides (NKCC1 to the basolateral side and NCC and NKCC2 to the apical side) and their nephron locations specified in writing next to each transporter protein. The reference list has been corrected and is now in a consistent style in keeping with the required format of the journal. Any additional or altered text due to the addition of more recent literature, minor corrections or better illustration of points are highlighted in yellow.

Reviewer 2 Report
In the present paper, the authors reviewed recent literature and explained pathogenesis of Gordon syndrome (GS). Although the author tried to include various theories, several key molecular mechanisms are missing or different from current consensus.
・In the section of “3. The regulation of NCC”, they describe that WNK4 negatively regulate NCC. In early 2000, It was controversial issue whether WNK4 regulate NCC negatively or positively. However, all review articles about WNK signaling which published in these 3 years describe WNK4 positively regulate NCC through SPAK/OSR1. Currently, it is generally accepted that WNK4 is a positive regulator of NCC. This is clearly supported by the observation that WNK4 knockout mice manifest minimal NCC activity. WNK1 activate NCC through SPAK/OSR1 not WNK4. WNK3 has a minor role in distal nephron in kidney. Overall, the references the authors site in this manuscript are old. The authors should site new studies about WNK signaling research and revise WNK regulation. And correct Figures accordingly.
・In section 4, the author discussed the interaction between SPAK/OSR1 and WNK kinases or CCC. However, their function is not discussed. In line 142-146 of “4. Discovery of SPAK and OSR1”, the physiological function of SPAK/OSR1 is ambiguous. SPAK/OSR1 lies downstream of WNKs and phosphorylates and activates NCC. The authors should describe it more clearly. They play important role in pathogenesis of GS especially because inactive mutation of SPAK/OSR1 has been experimentally proven to eliminate positive effect of WNK kinases on NCC. At least, the author must mention that WNK kinases phosphorylate and activate SPAK/OSR1, and that phosphorylated/activated SPAK/OSR1 phosphorylate NCC, leading to NCC activation.
・The authors described in figures that NCC regulations by WNK kinases happens in DCT2 cells. However, to our knowledge, there is no evidence that WNK signaling functions in DCT2 not DCT1. Grimm et al. reported that activation of SPAK kinase only in DCT1 was able to recapitulate GS phenotype.
・In section 5, the authors introduced several theories about pathogenesis of hyperkalemia in GS. While ENaC, ROMK might play a role, this is not consistent with the fact that HCTZ is able to correct hyperkalemia in GS as the author also mentioned. Therefore, the authors should not treat every theories equally, but focus on the mechanism why increased NCC activity causes hyperkalemia.
・Among monogenic hypertensive diseases, GS is the only disease which shows the phenotype of hypoaldosteronism. The authors should put more emphasis on this clinical characteristics. Also, a figure which explains why ENaC activation causes potassium loss in other monogenic hypertension should be included and explained, when general readership of the journal are taken into account.
・In Table 2, patients with GS manifest hypercalciuria, not hypocalciuria.
・In addition to Lifton's group, in 2012 French group also reported that KLHL3 mutation causes GS (Louis-Dit-Picard H et al. Nat Genet. 2012;44(4):456-460). This should be included.
・p.7 line 223: ENaC depression results in decreased Na reabsorption, instead of increased reabsorption.
・This manuscript lacks information about distribution along nephron segment of WNKs, SPAK/OSR1, Cullin-3 and KLHL3. The authors need to add this information to this review.
・In figure 1-3, the authors need to add information of apical side and basolateral side.
・There are the two following major mistakes in figure 3. 1. NCC and NKCC2 localizes to the apical membrane. 2. NKCC2 mainly expresses in TAL. The authors should revise this figure.
・In references, the format is not unified. There is under bar at journal name in some references. Publish year is put in a round bracket in some references. Some references are redundant.
Author Response

(The authors gave the same response as above.)

Round 2
Reviewer 2 Report
In the revised version of Manuscript, Mabillard et al. have referred to more recent literature including those about positive regulation of NCC by WNK4 and function of SPAK/OSR1. The reviewer agree with most of the changes made by the authors, although there are some comments critical for this review, as shown below. I hope my comments should improve the manuscript.
In page2, line86, Gordon syndrome manifests hypoaldosteronism phenotype because increased NaCl reabsorption in DCT decreases luminal sodium flow in more distal nephron and makes ENaC less functional. It is not due to low plasma aldosterone concentration, at all. This is a really important point to understand PHAII.
In line 166-167, the authors mentioned that when WNK4 was overexpressed in mice, WNK4 inhibit NCC. Wakabayashi, et al. reported that NCC phosphorylation was dramatically increased in WNK4 transgenic mice in expression level dependent-manner. They discussed the discrepancy between the phenotypes of the two transgenic mice[Ref 53 and Wakabayashi, et al. Cell Rep. 3:858-68, 2013. PMID: 23453970] might be due to WNK4 expression level. The authors should incorporate this study to this manuscript.
3. In section 6, the authors mentioned that Kelch-like 3 interact with WNK1 and WNK4. Gordon syndrome-causing KLHL3 mutations affect the binding of WNK1 and WNK4. They referred only cell experiment study about the interaction between Kelch-like 3 and WNK1. In fact, mice carrying KLHL3 disease-causing mutation showed that both WNK1 and WNK4 were increased in the kidney [Susa, et al. Hum Mol Genet.23:5052-60, 2014. PMID:24821705. Sasaki, et al.Mol Cell Biol. 37: pii: e00508-16 PMID:28052936]. However, figure 4 shows that only WNK4 binds to Kelch-like 3. The authors should add WNK1 to figure 4 and incorporate these studies' findings to this section.
Author Response
Response to reviewers comments
Reviewer 2:
Comments and Suggestions for Authors
In the revised version of Manuscript, Mabillard et al. have referred to more recent literature including those about positive regulation of NCC by WNK4 and function of SPAK/OSR1. The reviewer agree with most of the changes made by the authors, although there are some comments critical for this review, as shown below. I hope my comments should improve the manuscript.
In page2, line86, Gordon syndrome manifests hypoaldosteronism phenotype because increased NaCl reabsorption in DCT decreases luminal sodium flow in more distal nephron and makes ENaC less functional. It is not due to low plasma aldosterone concentration, at all. This is a really important point to understand PHAII.
We have corrected this and revised this section.
In line 166-167, the authors mentioned that when WNK4 was overexpressed in mice, WNK4 inhibit NCC. Wakabayashi, et al. reported that NCC phosphorylation was dramatically increased in WNK4 transgenic mice in expression level dependent-manner. They discussed the discrepancy between the phenotypes of the two transgenic mice[Ref 53 and Wakabayashi, et al. Cell Rep. 3:858-68, 2013. PMID: 23453970] might be due to WNK4 expression level. The authors should incorporate this study to this manuscript.
We have expanded this section and discussed the murine models and their difference in phenotypes and added new references as suggested.
3. In section 6, the authors mentioned that Kelch-like 3 interact with WNK1 and WNK4. Gordon syndrome-causing KLHL3 mutations affect the binding of WNK1 and WNK4. They referred only cell experiment study about the interaction between Kelch-like 3 and WNK1. In fact, mice carrying KLHL3 disease-causing mutation showed that both WNK1 and WNK4 were increased in the kidney [Susa, et al. Hum Mol Genet.23:5052-60, 2014. PMID:24821705. Sasaki, et al.Mol Cell Biol. 37: pii: e00508-16 PMID:28052936]. However, figure 4 shows that only WNK4 binds to Kelch-like 3. The authors should add WNK1 to figure 4 and incorporate these studies' findings to this section.
We have revised figure 4 and included a discussion of WNK1 and WNK4 in the main text as well as adding comments and new references as suggested.
